# The Cumulative Risk of Chemical and Nonchemical Exposures on Birth Outcomes in Healthy Women: The Fetal Growth Study

**DOI:** 10.3390/ijerph16193700

**Published:** 2019-10-01

**Authors:** Leah Zilversmit Pao, Emily W. Harville, Jeffrey K. Wickliffe, Arti Shankar, Pierre Buekens

**Affiliations:** 1Department of Epidemiology, School of Public Health and Tropical Medicine, Tulane University, New Orleans, LA 70112, USA; harville@tulane.edu (E.W.H.); pbuekens@tulane.edu (P.B.); 2Department of Global Environmental Health Sciences, School of Public Health and Tropical Medicine, Tulane University, New Orleans, LA 70112, USA; jwicklif@tulane.edu; 3Global Biostatistics and Data Science, Department of Global Environmental Health Sciences, School of Public Health and Tropical Medicine, Tulane University, New Orleans, LA 70112, USA; sarti@tulane.edu

**Keywords:** cumulative risk, chemical exposures, nonchemical exposures, perinatal health

## Abstract

Metals, stress, and sociodemographics are commonly studied separately for their effects on birth outcomes, yet often jointly contribute to adverse outcomes. This study analyzes two methods for measuring cumulative risk to understand how maternal chemical and nonchemical stressors may contribute to small for gestational age (SGA). SGA was calculated using sex-specific fetal growth curves for infants of pregnant mothers (*n* = 2562) enrolled in the National Institute of Child Health and Human Development (NICHD) Fetal Growth Study. The exposures (maternal lead, mercury, cadmium, Cohen’s perceived stress, Edinburgh depression scores, race/ethnicity, income, and education) were grouped into three domains: metals, psychosocial stress, and sociodemographics. In Method 1 we created cumulative risk scores using tertiles. Method 2 employed weighted quantile sum (WQS) regression. For each method, logistic models were built with three exposure domains individually and race/ethnicity, adjusting for age, parity, pregnancy weight gain, and marital status. The adjusted effect of overall cumulative risk with three domains, was also modeled using each method. Sociodemographics was the only exposure associated with SGA in unadjusted models ((odds ratio) OR: 1.35, 95% (confidence interval) CI: 1.08, 1.68). The three cumulative variables in adjusted models were not significant individually, but the overall index was associated with SGA (OR: 1.17, 95% CI: 1.02, 1.35). In the WQS model, only the sociodemographics domain was significantly associated with SGA. Sociodemographics tended to be the strongest risk factor for SGA in both risk score and WQS models.

## 1. Introduction

The Environmental Protection Agency (EPA) has defined cumulative risk as “the combined risks from aggregate exposures to multiple agents or stressors” [1]. While much of the scientific community, including the EPA, has advocated for the study of cumulative risk models [1,2,3,4], this is rarely done in epidemiological studies. When it is studied, researchers typically do not combine social and chemical exposures. Several studies have used various methodologies to study chemical mixtures [5], and some scientists have combined psychosocial stressors [6], but most researchers tend to construct separate models for chemical and nonchemical stressors. Recently, researchers have advocated for combining these exposures in order to have a more complete understanding of how exposures interact to affect outcomes [2,3,7,8]. Although researchers have attempted to model the combined effects of chemical and nonchemical exposures to study lead and stress on cognition [9]; benzene and stress on birthweight [10]; and lead, cadmium, and polychlorinated biphenyls on blood pressure [11], methods to study combined chemical and nonchemical are still lacking.

Studying multiple chemical and nonchemical exposures is of special interest to researchers studying perinatal health. One area of concern are indicators for fetal growth restriction, such as small for gestational age (SGA), which have shown to have both short-term and long-term effects on infants as they grow into adulthood [12]. Despite extensive research into the exposures contributing to SGA, understanding the exact etiology has proven difficult. Several studies reveal possible associations with SGA, including environmental toxicants, race, ethnicity, income, maternal educational attainment, and maternal mental health [13,14,15], but findings are often inconsistent and the biological rationale is not conclusively known. Studies are consistent in indicating that women who are non-Hispanic-black, low income, and have not attained degree past high school [15,16,17,18,19] are more at risk for most adverse perinatal outcomes, including SGA. 

Chemical exposures are also risk factors for reduced fetal growth. Heavy metals, which are common neurotoxins [20], have been linked to impeded fetal development at high levels of occupational exposure. In utero exposure to certain heavy metals, like lead (Pb), cadmium (Cd), and mercury (Hg), have been associated with low birthweight (LBW) and preterm birth (PTB) [21,22,23,24,25,26,27,28], although there are many studies that do not find an increased risk between chemical exposure and adverse birth outcomes [13]. This is similar to studies researching other chemicals. For example, researchers studying the effects of phthalates and pesticides on fetal development have reached inconclusive results [13,29,30]. However, there is evidence that, while at low doses there are no detectable risks when studied individually, there could be a cumulative risk when these low doses are studied in combination [31,32]. Animal models have indicated that if modeled cumulatively, these chemicals do in fact impact fetal development [33,34,35].

Beyond the possible link between environmental toxicants and birth outcomes, other researchers have focused on social stressors for their risk of adverse birth outcomes. Many point to the stress experienced during preconception and pregnancy, with evidence of an association between many indicators of stress and fetal growth [15]. Women who report having depressive symptoms tend to have worse birth outcomes [36,37], as well as a higher risk for SGA [14,38]. Although women who experience these psychosocial issues trend towards having adverse birth outcomes, like chemical exposures, literature reviews have reported some ambiguity; the evidence is far from conclusive that stress disorders directly cause adverse birth outcomes [37,39,40,41,42]. Inconclusive findings could be the result of inadequate methods for studying stress [43], but it could also be a result of the type of populations that are studied. When studying low risk, healthy pregnant women, Voegtline et al. found that stress indicators, such as depressive symptoms, trait anxiety, and emotional well-being, tended to be non-significantly correlated with cortisol measured throughout pregnancy [44]. 

Further complicating perinatal research is the possible interaction between chemical and nonchemical exposures. Common socioeconomic exposures, like race/ethnicity, income, and education, are correlated with each other [45,46]. African Americans, for example, have higher incidences of PTB, impaired fetal growth, and maternal mortality [47,48], with studies identifying stress induced by racism as an underlying factor [18,49,50]. Levels of cortisol are higher among women of lower socioeconomic status (SES) [51], with black women tending to report higher exposure to stress compared to other races [41]. Race and ethnicity are commonly multi-correlated with other social factors [45] and those that are disadvantaged are at additional risk to be ‘vulnerable’ to environmental exposures and psychosocial stress. DeFur et al. theorized that vulnerability, which is typically a result of low SES, interacts with physical and social environments and results in disparities in health outcomes [52]. This theory is supported by numerous studies finding that lower-SES populations most commonly live in areas with higher environmental exposures, including proximity to pollution, industrial areas, and other areas containing large amounts of environmental toxicants [7,53,54,55,56], and are thus more vulnerable to be exposed to psychosocial stress as a result of these circumstances [57,58]. SES may also be an indicator of different dietary intakes or occupational exposures that may put people of lower SES at greater risk for environmental toxicants [59].

Due to the multi-correlated nature of risk factors contributing to fetal growth, perinatal research could benefit from researching exposures cumulatively. This is especially true when studying relatively healthy women, as healthy women tend to be exposed to toxicants at levels lower than the regulatory concerned and associations of individual chemical and nonchemical exposures with birth outcomes are typically only evident at high levels of exposure [13]. However, just as in chemical exposures that may only show risk at low doses with multiple exposures, there could be associations between the nonchemical and chemical risk factors and perinatal outcomes, if a combination of lower levels of exposure are measured cumulatively. These risks also may interact with each other, causing a high degree of multicollinearity. We have included a conceptual model to explain the hypothesized interrelationships of possible chemical and nonchemical stressors (Figure 1). The objective of this study is to understand whether cumulative risk, indicated by the combination of exposure to heavy metals, psychosocial stress, and sociodemographic risk, contributes to incidence of SGA infants in a healthy cohort of pregnant women. We compare two possible ways of assessing this cumulative risk, one a relatively simple risk score that weights exposures equally, and one which allows for varying weights, weighted quantile sums [60,61].

## 2. Material and Methods

### 2.1. Study Participants

We conducted a secondary analysis on the data collected through the Fetal Growth Study, a cohort study conducted on pregnant women through the Eunice Kennedy Shriver National Institute of Child Health and Human Development (NICHD) with the primary goal of calculating a national standard for fetal growth and size for gestational age. Secondary goals were to study fetal growth in women with gestational diabetes mellitus (GDM) among the *n* = 468 obese participants, as well as measuring fetal growth in twin gestations. The study was conducted between July 2009 and January 2013 in 12 locations: Columbia University (NY), New York Hospital, Queens (NY), Christiana Care Health System (DE), Saint Peter’s University Hospital (NJ), Medical University of South Carolina (SC), University of Alabama (AL), Northwestern University (IL), Long Beach Memorial Medical Center (CA), University of California, Irvine (CA), Fountain Valley Hospital (CA), Women and Infants Hospital of Rhode Island (RI), and Tufts University (MA). The data are comprised of 3270 low-risk pregnant women with singleton gestation, with 468 of them classified as obese. Women were recruited within their first trimester, had a singleton gestation, did not consume alcohol, were nonsmokers, and were not suspected to have a fetus with a fetal abnormality. Alcohol and smoking habits were assessed upon recruitment in a questionnaire which asked participants the amount of alcohol they consumed the amount they smoked within 3 months prior to pregnancy and current consumption and smoking status. The dataset includes six visits: one initial recruitment visit and five additional follow-up visits. Participants were randomized into four groups (A, B, C, or D) with each group having varying schedules of follow-up visits, depending on gestational age. The first follow-up visit for group A was scheduled for 15 to 17 weeks gestation; in B, for 17 to 18 weeks gestation; in C, for 19 to 21 weeks gestation; and in D, for 21 to 23 weeks gestation. A detailed account of the cohort recruitment, including exclusion criteria, has been previously published [62]. For the purposes of this analysis, only women with a singleton gestation and who had a live birth were included in this study. After accounting for missing data, the complete dataset consisted of *n* = 1569 women (Figure 2).

### 2.2. Variables

Education, race/ethnicity, and income were assessed upon recruitment. Women were asked their highest degree obtained and whether they were white, black, American Indian, an Alaskan Native, or Asian. Women were additionally asked whether they were Hispanic. Women reported their family income from the previous year in 10 categorical groupings, which we collapsed into 4 categories for better power: (1) ≤$29,999, (2) $30–$49,999, (3) $50–$74,999, (4) ≥$75,000. We categorized education into high school or less, some college or an associate’s degree, and completed college. Race/ethnicity was categorized as black, white, Hispanic, or Asian. We collapsed the categories as black or non-black (non-black participants include white, Hispanic, and Asian) in the final analysis, because black women tended to be more at risk for adverse outcomes, while risk was similar in the other groups.

Information on depression and perceived stress was collected during the first and second trimesters across a total of seven study visits (recruitment and six follow-up visits); however, we only included information from the first 5 visits due to missing psychosocial information after five study visits. Each depression score and perceived stress score was averaged across study visits. Perceived life stress was assessed using the Cohen Perceived Stress scale. The 10-item questionnaire captures perceived stress in daily life [63,64], where the top tertile used as an indicator of high stress. The Edinburgh Postnatal Depression Index (EPDS) was used to assess depression [65], with a cutoff value of 12 used to indicate depression [66]. Birthweight, gestational age, and sex of the infant were assessed through birth record abstraction at the participating hospitals. 

The outcome variable was SGA. Infants were categorized as being SGA if their weight fell below the 10th percentile for their recorded gender and gestational age, in accordance with published national standards [67]. 

Women were dichotomized as being married/cohabitating with a partner or single. Parity was categorized into nulliparous, one previous birth, or two or more births. Age was categorized to 24 years old or less, 25–34 years old, and 35 or over based on the risk distribution within the age categories. We used the guidelines published by the Institute of Medicine (IOM) [68] to determine whether women’s weight gain during the entire pregnancy was adequate, under the recommended amount, or over the recommended amount according to reported pre-pregnancy BMI. 

### 2.3. Specimen Collection and Analysis

For this study, we analyzed data from blood samples collected during recruitment (8–12 weeks gestation). Obese women provided 30 mL non-fasting blood samples while women from other BMI groups provided 20 mL samples, which were sent to the Trace Elements Section of the Laboratory of Inorganic and Nuclear Chemistry at the Wadsworth Center, New York State Department of Health (Albany, NY, USA) where they were stored at −80 °C. Whole blood Pb, Cd, and Hg were determined by coupled plasma-mass spectrometry (ICP-MS). Methods to quantitate heavy metals were previously validated [69,70,71,72,73]. In total, 2063 participants provided blood samples upon recruitment, which were then analyzed for Pb, Cd, and Hg. Seven percent of participants who provided blood samples (*n* = 148) had amounts less than the limit of detection (LOD) for Pb; this amount was 1% (*n* = 23) for Hg and 8% (*n* = 171) for Cd. Values that were below the LOD were included during analysis in order to prevent introducing bias [70] using the values that were originally detected, consistent with other studies using the same laboratory analysis methods [71].

### 2.4. Statistical Analysis

Cumulative models were analyzed in two ways: (1) by creating a cumulative risk score, using tertiles of the exposure and (2) by constructing weighted quantile sum (WQS) regression models. Cumulative risk scores were created by first dividing each predictor into tertiles; the highest tertile was considered exposed (x = 1) while the 2 lowest tertiles were unexposed (x = 0). We used tertiles given their precedent in studies of chemical exposures [21]. In a sensitivity analysis using quartiles and quintiles we found that results did not vary widely from the tertile method; however, using tertiles avoided issues with model convergence. In the first method, we divided each of the continuous exposures (Pb, Cd, Hg, EPDS scores, and perceived stress scores) into tertiles, with the highest tertile considered exposed. The ordinal predictors (income levels, and educational attainment) were considered exposed if income was <$30,000 and if women had attained less than a high school education. We created indices for each of the three domains: (1) metals, (2) psychosocial stress, and (3) sociodemographics. Each domain’s index was equal to the sum of the variables within that domain. The total cumulative domain was also created, consisting of the sum of all three domains. We created an unadjusted regression model for each of the four domains individually (metals, psychosocial, sociodemographics and total cumulative), as well as an additional unadjusted logistic regression model that assessed metals, psychosocial stress, and sociodemographic in a single model. We then created adjusted models looking at the three domains models separately in three different models. Lastly, we created two fully adjusted models. The first included Metals Psychosocial stress, and sociodemographics adjusted for parity, marital status, weight gain during pregnancy, and age. The last model included the total cumulative domain adjusted for parity, marital status, weight gain during pregnancy, and age.

We compared the above models to those developed using the weighted quantile sum (WQS) regression method developed by Carrico et al. [60]. WQS regression allows researchers to create an index of correlated predictors that are weighted according to their strength of association with the outcome. Weaker variables are zeroed out in the WQS index. The method accounts for highly correlated exposures and was developed in the context of multiple chemical exposures, such as phthalates [60,61].

Specifically, the equation presented by Carrico et al. seeks to calculate the weights of *c* set of correlated variables:$$g\left(\mu \right)=\beta_{0}+\beta_{1}\left(\overset{c}{\underset{i=0}{\sum }}w_{i}q_{i}\right)+z^{\prime }\varphi$$

The sum term is the index for the *c* items, and weights are represented by the sum of w_i_. Each w_i_ is constrained to a value between 0 and 1. All confounders are represented by $z^{\prime }\varphi$. Prior to analysis, the data is split into two datasets at random: a training dataset and the validation dataset [60,61]. Using the training dataset, bootstrap samples are selected and the strength of the associations for each c item is determined by the beta coefficient. The index is calculated based on the mean *w_i_*s across all bootstrap samples. Weights are estimated based on optimization algorithms employed to maximize the likelihood in a nonlinear model. Variables with more influence within the quantile were assigned higher weights. Variables are selected based on a previously set significance threshold (we used *p* = 0.10).

Four models were created, one for each domain (metals, psychosocial, and sociodemographic) and one for the total cumulative index that included all domains. For each domain, we created weights adjusting for race/ethnicity and confounders (weight gain, parity, marital status, and age). The test dataset consisted of a random sample of 40% of the observations with the results validated in the remaining 60% of the sample population. Weights in the unadjusted and adjusted model (each of the domains modeled separately with race/ethnicity and confounders) were noted, as well as the final betas (β) for each of the models. The following equations represent the WQS models used:*logit (SGA)* = β_0_+ β_Metals_*x*_Metals_ + ε

*logit (SGA)* = β_0_+ β_Psychosocial_*x*_Psychosocial_ + ε

*logit (SGA)* = β_0_+ β_Demographics_*x*_Demographics_ + ε

*logit (SGA)* = β_0_+ β_Total_*x*_Total_ + ε.

*logit (SGA)* = β_0_+ β_Total_*x*_Total_ + β_Race_*x*_Race_ + β_Age_*x*_Age_ + ε

SAS software 9.4 (SAS Institute Inc., Cary, NC, USA) was used to generate the descriptive and cumulative risk scores. R software was used to find weights and estimates in WQS regression analysis [74]. All participating sites received human subjects’ approval and participants gave their informed consent before data was collected. As a secondary analysis of de-identified data, this analysis was not considered human-subjects research.

## 3. Results

There were 2038 participants that met the inclusion criteria and provided a blood specimen. Women were divided evenly between non-Hispanic white, non-Hispanic black, and Hispanic (27.3%, 26.1%, and 27.4% respectively) (Table 1). The majority of women either earned <$30,000 (*n* = 489, 27.8%) or ≥$100,000 (*n* = 521, 29.6%). Only 3% of women (*n* = 58) scored high enough on the EPDS to be categorized as depressed. Seven percent of infants (*n* = 150) were SGA. 

A description of continuous variables is presented in Table 1. About 7% (*n =* 148) of women had Pb levels below the limit of detection. Only 1% (*n =* 23) of women had Hg levels below the limit of detection and 8% (*n =* 171) of women had Cd levels below the limit of detection. The average Pb blood level was 0.51 micrograms per deciliter (μg/dL) and only 1% (*n =* 21) of the study population had a Pb blood level of greater than 5 μg/dL. Only one participant had an excessive level of Cd, measured at 5 μg/L, and one participant had blood Hg content (>5 μg/L). 

Most variables were individually associated with SGA (Table 2), with the exception of depression (OR: 0.32, 95% CI: 0.08, 1.33) and Hg (OR: 0.91, 95% CI: 0.78, 1.45). Non-Hispanic black women were at a higher risk of having SGA infants compared to non-black women (OR: 2.46, 95% CI: 1.59, 3.81). Higher levels of Cd and Pb also were associated with having an SGA infant (OR: 1.18, 95% CI: 1.04, 1.35 and OR: 1.22, 95% CI: 1.03, 1.35). Higher income was protective against SGA when compared to lower income (less than $30,000) and attaining higher levels of education was also protective against SGA. Women with previous pregnancies, married women, and women who were older than 25 were less likely to have SGA infants compared to nulliparous, nonmarried, and younger women respectively. 

Variables within domains tended to be correlated with each other (Table 3); however, Hg and Pb were not significantly correlated. Though many of the variables were statistically significantly correlated, education and income appeared to be the only variables highly correlated to each other (r = 0.68, *p* < 0.05). Depression and perceived stress were moderately correlated (r = 0.31, *p* < 0.05). Only the two sociodemographic variables (education and income) were correlated with SGA (not included in table).

In unadjusted cumulative models (Table 4), the demographic index and the total cumulative index were the only variables associated with SGA. The total cumulative index (Table 4) was associated with SGA even with race/ethnicity included in the model (OR: 1.17, 95% CI: 1.02, 1.35). While not significantly associated with SGA, the unadjusted metals domain and the psychosocial domain appeared to have an association, given that the lower confidence level was close to the null (OR 1.16, 95% CI 0.97, 1.38 and OR: 1.20, 95% CI 0.96, 1.52 respectively). In the adjusted models, inputting the domains separately did not yield any associations between the indices or race/ethnicity and SGA.

The weights for the WQS regression models and the betas are described in Table 5. The sociodemographic domain was the only domain associated with SGA (β = 0.41, SE = 0.20, *p* = 0.04). According to the weights assigned in the WQS regression models, income appears to have the most influence on the index, as the weight for income was 0.69, while it was only 0.31 for education (Table 5). Weights for adjusted and unadjusted WQS models, betas (β) and standard errors (SE) for final adjusted WQS regression models n WQS models, Fetal Growth Study.

## 4. Discussion

This analysis indicates that a cumulative measure of multiple exposures could be associated with SGA. The index that included all three domains in the cumulative risk score analysis (heavy metals, psychosocial stress, and sociodemographics) was consistently associated with SGA, suggesting that modeling exposures cumulatively could help explain how lower levels of exposures jointly increase women’s risk of these adverse birth outcomes. However, in WQS regression models, the total cumulative model was not associated with SGA, and the only domain to be associated with the outcome was the sociodemographic index. Sociodemographics and race/ethnicity were so strongly associated with the outcome they outweighed effects of other exposures. Previous studies have indicated that controlling for indicators of socio-economic levels could control for other effects. This results in only the strongest SES terms being associated with the outcome, while producing null results among other indicators. In Lefmann et al’s study, for example, they believed that the reason why Medicaid status was the only predictor associated with SGA, LBW, and PTB was because Medicaid controlled for other correlated indicators, such as race and experiencing stressful events [75]. They postulated that Medicaid status could be an indicator for poverty, thereby negating associations between other sociodemographic and stress variables in the models. A similar process could be operating with respect to exposures to toxicants; certain lower-SES populations are more vulnerable to environmental exposures [52]. Several studies have indicated disadvantaged populations are at higher risk of exposure to both chemical and nonchemical exposures [3,76]. DeFur et al., described the vulnerability in lower SES groups, meaning not just a susceptibility of lower SES groups to harmful exposures, but also an inability of these groups to recover. That results in disparities in outcomes between SES groups [52]. DeFur et al.’s conclusion is plausible given that lower SES people tend to live in closer proximity to industrial areas and other areas with higher chemical exposures [7,53,54,55,56]. However, depending on the population, other potential sources of pollutant exposure (diet, consumer products, and occupation) may not have the same correlation with SES: for instance, jobs with relatively high levels of exposure may also be high-paying, or women with higher incomes may be able to afford seafood that contains mercury. While the study does not allow us to determine why lower SES women had a higher risk of having SGA infants, the analyses do provide more evidence on the lasting and overwhelming impacts of lower income and educational attainment on perinatal outcomes.

WQS regression’s ability to weight predictors may be of special interest to researchers who study multiple correlated predictors, as it avoids a simplistic assumption of equal weighting provided by a simple cumulative risk score, such as our initial analysis provided. It should be noted, however, that WQS still assumes an equal increase/decrease in risk with every one-quantile change, which may be incorrect if an association is nonlinear. We found no other studies that used the WQS method to estimate the joint effect of both chemical and nonchemical exposures. Since its development, the approach has been primarily used to combine multiple chemical exposures [61,77,78,79,80,81], although more recently, the approach has been used to create SES indices [82], nutrient indices from self-reported food frequency questionnaires [11,83,84], and an index for psychosocial stress [6]. Though the authors of these studies chose to build models that did not combine chemical and nonchemical exposures, some authors did attempt to build WQS coefficients that combined several domains. Yorita, Christensen et al. created three WQS indices (heavy metals, co-planar polychlorinated biphenyls [PCBs], and non-dioxin-like PCBs) to predict alanine amino transferase (ALT) levels (which indicate liver damage). Czarnota et al. used one WQS index to estimate the combined effect of PCBs, polycyclic aromatic hydrocarbons (PAH), and pesticides [77]. Risk factors across different domains are commonly correlated with each other, so using an approach that accounts for this correlation could be helpful for researchers in many fields of epidemiology. However, if exposure variables are strongly correlated with each other and weakly correlated with the outcome WQS will perform poorly [60].

### Limitations

We conducted this analysis on a cohort of pregnant women with no previous history of very PTB or LBW infants, as well as no other pregnancy complications. Results cannot be generalized to other populations of women and are not representative of the entire population of the United States. A review of literature reveals that scientists are also currently unaware of the amounts of heavy metals that may cause adverse birth outcomes. While most women did not have blood samples that were higher than the CDC recommended limits, there is still evidence that lower levels of exposure could have adverse effects on perinatal outcomes (which was one of the objectives of this study) [25,33,34]. As no clear cut-off point was available from the literature, we used the highest tertile to classify exposure. 

The primary intention of this study was to understand whether cumulative risk, of chemical and nonchemical exposures, was associated with SGA in a healthy population of women, which included women who were obese but otherwise healthy. Investigating women who do not present prior health issues results in the exclusion of higher-risk women. If the examined risk factors are associated with intermediates, selection bias could exist; regardless, this fact limits generalizability. Studying the effects in more-exposed and/or less-healthy populations is an important next step in this research. While we chose variables that have effects on fetal growth, there may have been other variables that could be more appropriate for this analysis that were not measured for this study. Other chemical exposures, such as PCBs, air pollution, or other environmental pollutants, may interact with heavy metals, but cannot be studied in this secondary data analysis [11,85,86]. We were similarly limited to the psychosocial stress measures studied in this analysis; we were unable to study other indicators of stress, such as childhood traumatic events or post-traumatic stress disorder (PTSD), which have also been associated with adverse birth outcomes [44,87]. Each factor studied is affected by a degree of measurement error; some of them, such as stress and metal levels, may change over time as well. (Age is likely to be a susceptibility factor, but thoroughly examining this interaction would require multiple measures over time.) Combining multiple imperfectly measured variables can attenuate effects under some circumstances [88]. As methods for combined exposure analysis develop, the degree to which errors are amplified will need to be assessed. The study also suffers from missing data, as illustrated in Figure 1. One limitation of WQS, or any assessment studying cumulative risk, is that the many exposure variables included in the study will inevitably lead to more missing data compared to traditional studies, which only assess one exposure variable with one outcome. To the extent that missing data is informative or associated with exposure and outcome, this may bias the sample; most likely, women with complete data will be at lower risk than the overall cohort. 

## 5. Conclusions

This study demonstrates that risk factors may affect SGA cumulatively and introduces WQS as a possible tool to measure cumulative risk. WQS has been presented as a possible tool to measure risks in perinatal research, as well as presenting the possibility of using the method in studies involving chemical and nonchemical risk factors. The present analysis also supports previous research on the importance of socioeconomic factors on reproductive health outcomes. When researchers study cumulative risk, special attention should be placed on acknowledging the important role of sociodemographics on maternal health.

## Figures and Tables

**Figure 1 ijerph-16-03700-f001:**
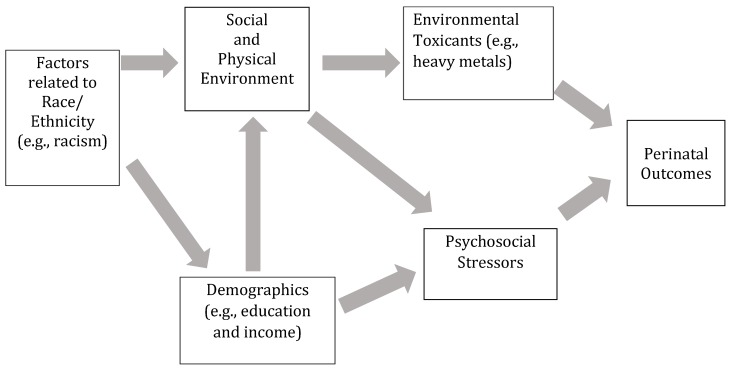
Conceptual model of the relationship between chemical and nonchemical exposures on perinatal outcomes.

**Figure 2 ijerph-16-03700-f002:**
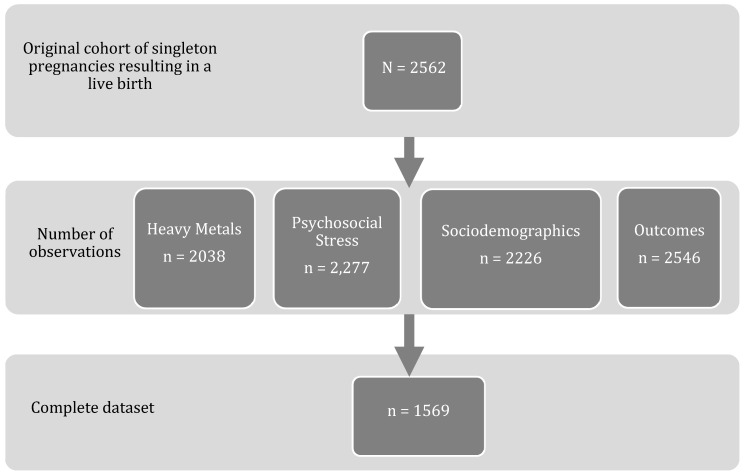
Missing data in the Fetal Growth Study.

**Table 1 ijerph-16-03700-t001:** Population characteristics, The Fetal Growth Study (*n* = 2038).

Participant Characteristic		*n*	%
Race			
	Non-Hispanic White	557	27.33
	Non-Hispanic Black	532	26.10
	Hispanic	559	27.43
	Asian & Pacific Islander	390	19.14
Income			
	Less than $30,000	489	27.78
	$30,000–$39,999	155	8.81
	$40,000–$49,999	138	7.84
	$50,000–$74,999	216	12.27
	$75,000–$99,999	241	13.69
	$100,000 or more	521	29.60
Education			
	Less than high school	208	10.21
	High school diploma or GED or equivalent	350	17.17
	Some college or Associate degree	593	29.10
	Bachelor’s degree	509	24.98
	Master’s degree or Advanced degree	378	18.55
Perceived Stress			
	No (<75tth percentile)	1433	78.82
	Yes (≥75th Percentile)	385	21.18
Depression			
	No	1980	97.15
	Yes	58	2.85
SGA			
	No	1875	92.59
	Yes	150	7.41
Preterm Birth			
	No	1902	93.74
	Yes	127	6.26
Low Birthweight			
	No	1921	94.86
	Yes	104	5.14
Weight gain			
	Adequate	566	30.64
	Under	317	17.16
	Over	964	52.19
Parity			
	0	1006	49.36
	1	696	34.15
	2+	336	16.49
Marital Status			
	Not married or cohabitating	490	24.07
	Married or cohabitating	1546	75.93
Age			
	≤24	583	28.61
	25–35	1171	57.46
	>35	284	13.94
Variables Measured Continuously			
	Mean (SD)	Median	25^th^ and 75^th^ percentiles
Perceived Stress	28.51 (9.09)	28.00	26.00, 30.00
Depression	4.67 (3.38)	4.00	2.00, 7.00
Weight gain (kg)	15.31(5.99)	14.97	11.79, 18.60
Age	28.21 (4.7)	29	24.00, 32.00
Pb (µg/dL) ^1,2^	0.51 (5.22)	0.11	0.06, 0.22
Cd (µg/L) ^1,3^	0.03 (0.40)	0.01	0.01, 0.02
Hg (µg/L) ^1,4^	0.32 (0.37)	0.22	0.11, 0.43

^1^ Heavy metals presented eliminated observations below the limits of detection (LOD). ^2^
*n =* 2038, ^3^
*n =* 2063, ^4^
*n =* 2063.

**Table 2 ijerph-16-03700-t002:** Bivariate analysis comparing exposures and confounders to small for gestational age (SGA), the Fetal Growth Study.

Participant Characteristic	OR	95% CI	*p*-Value
Race			0.0003
Not Black	Ref		
Non-Hispanic Black	2.46	1.59, 3.81	
Income			
Less than <$30,000	Ref		0.01
$30,000–$39,999	1.38	0.82, 2.34	
$40,000–$49,999	1.06	0.69, 1.88	
$50,000–$74,999	0.94	0.56, 1.59	
≥$75,000	0.57	0.37, 0.86	
Education			
HS or Less	Ref		
Some college or Associate degree	0.81	0.57, 1.17	0.002
At least college degree	0.53	0.36, 0.76	
Log Pb	1.18	1.04, 1.35	0.01
Log Cd	1.22	1.03, 1.35	0.01
Log Hg	0.91	0.78, 1.45	
Perceived Stress (75^th^ percentile)	1.04	0.70, 1.07	0.02
Depression	0.32	0.08, 1.33	0.10
Weight Gain			
adequate	Ref		<0.0001
under	1.56	1.05, 2.33	
over	0.56	0.33, 0.81	
Parity			
0	Ref		
1	0.61	0.43, 0.86	0.004
2+	0.59	0.38, 0.91	
Married vs. Nonmarried	0.61	0.44, 0.83	0.002
Age			
≤24	Ref		<0.0001
25–35	0.52	0.38, 0.72	
>35	0.46	0.28, 0.78	

**Table 3 ijerph-16-03700-t003:** Spearman correlation coefficients for tertiles of heavy metals, psychosocial stress, and categories of sociodemographics.

Variables	Pb	Cd	Hg	Income ^1^	Education ^1^	Perceived Stress	Depression
**Pb**		0.22	0.01	0.08	0.05	0.00	0.02
*p*-value		<0.0001	0.53	0.01	0.02	0.93	0.29
**Cd**			0.09	0.02	0.00	−0.02	0.05
*p*-value			<0.0001	0.24	0.82	0.42	0.03
**Hg**				−0.13	−0.12	−0.01	0.05
*p*-value				<0.0001	<0.0001	0.63	0.03
**Income**					0.68	−0.02	0.16
*p*-value					<0.0001	0.42	<0.0001
**Education**						−0.07	0.10
*p*-value						0.002	<0.0001
**Perceived Stress**							0.31
*p*-value							<0.0001
**Depression**							
*p*-value							

**^1^** Income and education were in reverse order (highest income and education to lowest).

**Table 4 ijerph-16-03700-t004:** Unadjusted and adjusted associations between composite variables for three domains (metals, psychosocial stress, and demographics) and SGA, the Fetal Growth Study *.

Domain	Unadjusted Models Using Highest Tertile as Exposed		Unadjusted Individual Domains, without Race/Ethnicity Included		Adjusted Models with All Individual Domains as Exposures	
	OR	CI	OR	CI	OR	CI
Metals	1.16	0.97, 1.38	1.18	0.95, 1.46	1.17	0.93, 1.48
Psychosocial	1.20	0.96, 1.52	1.16	0.90, 1.50	1.24	0.95, 1.62
Demographic	1.35	1.08, 1.68 *	1.29	1.01, 1.65 *	1.10	0.80, 1.50
Total Cumulative	1.21	1.06, 1.37 *	-	-	1.17	1.02, 1.35 *

* Significant at <0.05 level.

**Table 5 ijerph-16-03700-t005:** Weights for adjusted and unadjusted weighted quantile sum (WQS) models, betas (β), and standard errors (SE) for final adjusted WQS regression models, Fetal Growth Study.

Domain	Variable	Unadjusted	Adjusted	β (SE)	*p*-Value
Metals	Pb	0.49	0.48	0.30 (0.18)	0.10
	Cd	0.17	0.52		
	Hg	0.34	0.00		
Psychosocial	Depression	0.82	0.67	0.04 (0.19)	0.85
	Perceived Stress	0.18	0.33		
Sociodemographic	Income	0.69	0.12	0.41 (0.20)	0.04
	Education	0.31	0.88		
Cumulative WQS variable	Pb	0.28	0.03	0.37 (0.38)	0.33
	Cd	0.07	0.30		
	Hg	0.01	0.06		
	Depression	0.15	0.14		
	Perceived Stress	0.14	0.15		
	Income	0.20	0.22		
	Education	0.14	0.10		

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
