# Peer review of "The Cumulative Risk of Chemical and Nonchemical Exposures on Birth Outcomes in Healthy Women: The Fetal Growth Study"

_ijerph, 2019, doi:10.3390/ijerph16193700_

Round 1

Reviewer 1 Report

Comments and Suggestions for Authors:

Overall comments: I think the authors are doing interesting work by using a cumulative risk approach to better understand risk for SGA.

Abstract:

Line 18: Spell out “2”. All numbers less than less than 10 should be spelled out. Line 21: What does NICHD stand for? Lines 23, 27, 30: Spell out “3”.

Introduction:

Line 40: Comma after “studied”. Lines 42 and 43: I think you should say “…but fewer researchers tend to construct separate models for chemical and nonchemical stressors.” I would cite a couple of papers that do because there are studies that combine chemical and nonchemical stressors into one model. Line 59: “has” should be “have”. Lines 116, 117 and 128: Spell all of the numbers out that are less than 10. Line 129: Income categories should be number, 1)..2)…3) and 4). Line 131: Did you forget to include American Indian or Alaskan Native here? Lines 135 and 136: Spell out the numbers. Line 139: …“with” should be “where”. Add “is” after tertile. Line 147: Spell out the numbers less than 10. Lines 153 and 154: Why did the obese and normal weight participants provide different blood sample (20 ml vs. 30 ml). Lines 168 to 172: The sentence beginning with “We used tertiles…” is too long and should be broken down. Lines 171 and 176: Spell the numbers out. Line 176 and 177: Number the domains: 1)…2)…3). Line 178, 179, 182 and 183: Spell the numbers out. Line 196: Comma after “analysis”. Line 197: …”is” Line 197, 202 and 203: Spell out the numbers.

Results

Table 1: This table should fit on one page. Left align the characteristic headings and indent the information beneath 5 spaces, for example:

Race

     Non-Hispanic White

     Non-Hispanic Black

     Hispanic

     Asian & Pacific Islander

Income

     Less than $30,000

     $30,000 - $39,000

Note: The example table information above should should be single spaced. The formatting messed up after I pasted my comments in the text box.

Also, why do you use Pacific Islander in the table but not in the rest of the paper? Table 2: Format the same as Table 1. Line 243: Spell out number. Line 244: Add a period at the end of the sentence. Table 4: Spell out the number. Lines 275 to 278: These lines are a repeat of lines 263 to 266.

Discussion

Line 285: Spell out the number. Line 291: Add “that” after the word outcome. Lines 313 and 316: Spell out the numbers.

Conclusion

The conclusion seems a little weak. I would add a couple more sentence about the implications of your findings.

Author Response

Dear Reviewer,

We have carefully reviewed your comments. and are submitting the following responses:

Abstract:

Line 18: Spell out “2”. All numbers less than less than 10 should be spelled out. Line 21: What does NICHD stand for? Lines 23, 27, 30: Spell out “3”.

-All number less than 10 have been spelled out.

Introduction:

Line 40: Comma after “studied”. Lines 42 and 43: I think you should say “…but fewer researchers tend to construct separate models for chemical and nonchemical stressors.” I would cite a couple of papers that do because there are studies that combine chemical and nonchemical stressors into one model.

- We have added additional studies

Line 59: “has” should be “have”. Lines 116, 117 and 128: Spell all of the numbers out that are less than 10. Line 129: Income categories should be number, 1)..2)…3) and 4). Line 131: Did you forget to include American Indian or Alaskan Native here?

-These groups were not recruited in the study

Lines 135 and 136: Spell out the numbers. Line 139: …“with” should be “where”. Add “is” after tertile. Line 147: Spell out the numbers less than 10. Lines 153 and 154: Why did the obese and normal weight participants provide different blood sample (20 ml vs. 30 ml).

-This was a secondary data analysis and we believe the PI may have asked for additional biospecimens for an additional case-cohort study on GDM. However, we were unable to receive confirmation of this intention prior to submission deadline.

Lines 168 to 172: The sentence beginning with “We used tertiles…” is too long and should be broken down. Lines 171 and 176: Spell the numbers out. Line 176 and 177: Number the domains: 1)…2)…3). Line 178, 179, 182 and 183: Spell the numbers out. Line 196: Comma after “analysis”. Line 197: …”is” Line 197, 202 and 203: Spell out the numbers.

Results

Table 1: This table should fit on one page. Left align the characteristic headings and indent the information beneath 5 spaces, for example:

-Table 1 is revised back, It may have been reformatted during the editing process.

Race

     Non-Hispanic White

     Non-Hispanic Black

     Hispanic

     Asian & Pacific Islander

Income

     Less than $30,000

     $30,000 - $39,000

Note: The example table information above should should be single spaced. The formatting messed up after I pasted my comments in the text box.

Also, why do you use Pacific Islander in the table but not in the rest of the paper? Table 2: Format the same as Table 1.

-We combined non-Hispanic/not Black to one category vs. Black. When we looked at the risks associated with SGA, all non-black groups had similar associations, therefore dichotomized this variable

Line 243: Spell out number. Line 244: Add a period at the end of the sentence. Table 4: Spell out the number. Lines 275 to 278: These lines are a repeat of lines 263 to 266.

-It has been corrected

Discussion

Line 285: Spell out the number. Line 291: Add “that” after the word outcome. Lines 313 and 316: Spell out the numbers.

Conclusion

The conclusion seems a little weak. I would add a couple more sentence about the implications of your findings.

-We have added the following to the conclusion: “This study demonstrates that risk factors may affect SGA cumulatively and introduces WQS as a possible tool to measure cumulative risk. WQS has been presented as a possible tool to measure risks in perinatal research, as well as presenting the possibility of using the method in studies involving chemical and nonchemical risk factors. The present analysis also supports previous research in the importance of socioeconomic factors on reproductive health outcomes. When researchers study cumulative risk, special attention should be placed on acknowledging the important role of sociodemographics on maternal health.”

Reviewer 2 Report

This study uses a cumulative risk model framework to study the joint association of chemical, socioeconomic, stress-related exposures on the risk for delivering small for gestational age infants in The Fetal Growth Study. The goal of the study is to assess “cumulative risk” of exposure to heavy metals, psychosocial stress, and socioeconomic position on risk of SGA. The chemical exposures were metals measured in maternal blood and included lead, mercury, and cadmium. The stress measures were Cohen’s perceived stress scale and Edinburgh depression scores. Measures of socioeconomic position were income and education. Two methods of analyzing a cumulative risk score were used to summarize the joint impact of these three domains of exposure. The first created tertiles of exposure, and the second used weighted quantile sum (WQS) regression. The analysis found a significant association of income and education on risk for SGA, but not heavy metals or psychosocial stress. The study addresses an important question, but several considerations need further attention and limit the interpretability of the results.

Major comments:

There is a lack of attention paid to the potential time ordering and relationships between the variables in the analysis. For example, income and education are considered as independent exposures from stress and chemical exposures, but income and education can shape exposure to both stress and chemicals (and are related to one another, too). Variables like weight gain during pregnancy are likely on the causal pathway, but are controlled for as confounders. It is recommended to include a directed acyclic graph, which will help clarify how the authors are conceptualizing the relationships between the variables in the study, which will have implications for how the exposures are modeled in relation to one another and will inform what covariates should be controlled. The rationale for focusing on healthy women is not well-motivated, and the authors should spend more time examining the potential implications of this restriction. For example, if the exposures could influence development of potential health conditions that would disqualify women from participating, then there could be bias introduced if these conditions were also related to the outcome, because by conditioning on being healthy the authors would be conditioning on an intermediate of the exposure-outcome relationship. Furthermore, if there were an unmeasured confounder of these health conditions and the outcome, then by conditioning on health status, selection bias would be introduced, as health status would then be a collider of the unmeasured confounder and exposures. See the following for more discussion of these ideas: Cole SR, Platt RW, Schisterman EF, Chu H, Westreich D, Richardson D, Poole C. Illustrating bias due to conditioning on a collider. International journal of epidemiology. 2009 Nov 19;39(2):417-20. Hernán MA, Hernández-Díaz S, Robins JM. A structural approach to selection bias. Epidemiology. 2004 Sep 1;15(5):615-25. VanderWeele TJ, Mumford SL, Schisterman EF. Conditioning on intermediates in perinatal epidemiology. Epidemiology. 2012 Jan;23(1):1.

The choice of statistical approaches needs to be better described and defended, and additional limitations need to be acknowledged. For example, while the authors acknowledge that there may be interactions and/or correlations between exposures, their first method, based on creating a cumulative risk score, does not allow for them to be explicitly modeled (as far as I can tell). While the demonstration of the WQS regression method is interesting and appropriate, more discussion of its limitations is warranted. For example, the WQS regression method underweights variables that have high pairwise correlation. In particular, the method can breakdown when there is high correlation between exposures that dominate the correlation with the outcome. This seems like it may be a possibility in this study, and the authors should consider how this may affect their findings. In addition, the method assumes a constant change in risk between the quantiles, and assumes the effects of all exposures operate in the same direction. The authors should improve their discussion of the limitations of the method, and acknowledge how the method’s limitations may affect their conclusions.

Minor comments:

Introduction

The authors identify race as a risk factor for SGA; however, much evidence and theory points to racism, rather than race, as the mechanism that underlies documented associations between race and many adverse health outcomes. Historical policies and cultural forces that enforced economic exclusion and marginalization by race/ethnicity are also relevant when discussing the correlations between low income and educational attainment and race/ethnicity. These are important distinctions that the authors do acknowledge briefly but should mention each time they discuss race/ethnicity as an “exposure” or allude to the correlation of race/ethnicity with other risk factors. For example, the authors say in the introduction “Several possible risk factors for SGA have been identified, and include environmental toxicants, race, ethnicity, income, maternal educational attainment and maternal mental health…”. Instead they might consider “Several possible risk factors for SGA have been identified, and include environmental toxicants, structural and interpersonal racism, income, maternal educational attainment and maternal mental health…” See the following for more discussion on this topic: Krieger N. Does racism harm health? Did child abuse exist before 1962? On explicit questions, critical science, and current controversies: an ecosocial perspective. American journal of public health. 2003 Feb;93(2):194-9. Dominguez TP. Adverse birth outcomes in African American women: the social context of persistent reproductive disadvantage. Social work in public health. 2011 Jan 1;26(1):3-16.

Materials and methods

The authors claim that women were recruited if they did not consume alcohol and were nonsmokers. What is the relevant timeframe for these behaviors? Women who have never consumed alcohol and have never smoked are likely different from women who did not consume alcohol or smoke during pregnancy or in the year before birth. The authors should include a reference to the questionnaire that was used to evaluate these behaviors. They should also include a figure that illustrates how the final sample size was achieved. The authors cite a cohort profile that presumably goes into more detail about the study population and measures, but some additional information should be included in the manuscript. For example, are all the visits during pregnancy? The authors state there are 6 visits in the Study participants section, but then allude to 7 visits in the Variables section. Do the 5 study visits included in this study include the recruitment visit or not? When were maternal blood samples taken? A flow diagram of the study visits and timing of each measurement would be helpful. What are the 12 locations where women were recruited? The effect of income on health may be very different depending on cost of living which varies widely by city and region in the US. The authors claim they included values that were below the LOD in the analysis to prevent introducing bias. How was this done? This should be included in the methods. There are details relevant to the statistical analysis that are not clearly described enough. For example, the authors sum indicators of exposure to create a cumulative risk score, and then divide that sum into tertiles, with the highest tertile considered “exposed”. This would then assume each exposure has equal weight, which I am not sure makes sense. The authors should discuss the implications of this and/or include it as a limitation. Was there any missing data and/or loss to follow-up in the study, and if so, how was that handled? A description of this should be included in the methods. For the WQS regression, did the authors use p=0.10 as their significance threshold, or is that just an example? If it was an example, what was the threshold they used? Did they test the sensitivity of their results to this threshold? What does it mean that the “weights are based on the quantiles of the scales”? What are the scales? Can the authors give more information as to how they “created weights adjusting for race/ethnicity and confounders”? Were these weights the weights used in the WQS regression?

Results

The paragraphs on either side of Table 4 on page 9 are repeated. The results from the analysis using tertiles presented in Table 4 are not discussed in the text. The authors say that “the weight for education was 0.96 while it was only 0.04 for income.” In Table 5, however, the weights are 0.88 for education and 0.12 for income. I am not sure where the numbers in the text come from. We caution the authors to rely heavily on statistical significance when interpreting their results. For example, in Tables 4 and 5 there are several point estimates of similar size, but only a few confidence intervals where the lower CI is below 1. Recent guidance from statisticians strongly proposes that these differences be discussed with more nuance rather than focusing solely on the results whose confidence intervals do not cross the null or where the p-values are less than 0.05. For discussion of this point of view, see the following commentary: https://www.nature.com/articles/d41586-019-00857-9?fbclid=IwAR1jzbGpWu9wsHIwBdOu3byOielCLEQxPZMvHJ-3X4GW2gvy4eD98a7a9EU

Discussion

It is not clear to me how this study “helps explain how lower levels of exposures jointly increase women’s risk for adverse birth outcomes”. It just seems like this index was dominated by the socioeconomic indicators, and the stress and chemical exposures had no relationship with SGA separately or jointly. Did the authors consider analyzing the stress and chemical exposures jointly, while adjusting for income and education? Or since income and education may influence stress and chemical exposures, did they examine whether they could observe those relationships in their study, beyond just looking at bivariate correlations?

Author Response

Dear Reviewer,

Thank you for your comments. We have provided our responses below.

Major comments:

There is a lack of attention paid to the potential time ordering and relationships between the variables in the analysis. For example, income and education are considered as independent exposures from stress and chemical exposures, but income and education can shape exposure to both stress and chemicals (and are related to one another, too). Variables like weight gain during pregnancy are likely on the causal pathway, but are controlled for as confounders. It is recommended to include a directed acyclic graph, which will help clarify how the authors are conceptualizing the relationships between the variables in the study, which will have implications for how the exposures are modeled in relation to one another and will inform what covariates should be controlled.

-All the concerns the reviewers raises are the purpose of the study. With social, psychosocial, and environmental factors intimately correlated and with bidirectional causal relationships among them, the best method for analyzing effects is not at all clear. Multiple, equally plausible causal models could be specified and in many cases in this study and others a truly complete causal model would require a level of detail in measurement unlikely ever to be achieved (weight gain during pregnancy might be the result of dietary choices, which are constrained by medical status, culture, income, and nutritional knowledge, and the degree to which those choices are operating on any given day would vary – and pregnancy has 280 days). The study thus considers different methods to address the issue of multiple, tightly-related exposures, in part to determine whether these methods provide a better picture than more standard methods. We have also included a conceptual model in the paper.

The rationale for focusing on healthy women is not well-motivated, and the authors should spend more time examining the potential implications of this restriction. For example, if the exposures could influence development of potential health conditions that would disqualify women from participating, then there could be bias introduced if these conditions were also related to the outcome, because by conditioning on being healthy the authors would be conditioning on an intermediate of the exposure-outcome relationship. Furthermore, if there were an unmeasured confounder of these health conditions and the outcome, then by conditioning on health status, selection bias would be introduced, as health status would then be a collider of the unmeasured confounder and exposures. See the following for more discussion of these ideas: Cole SR, Platt RW, Schisterman EF, Chu H, Westreich D, Richardson D, Poole C. Illustrating bias due to conditioning on a collider. International journal of epidemiology. 2009 Nov 19;39(2):417-20. Hernán MA, Hernández-Díaz S, Robins JM. A structural approach to selection bias. Epidemiology. 2004 Sep 1;15(5):615-25. VanderWeele TJ, Mumford SL, Schisterman EF. Conditioning on intermediates in perinatal epidemiology. Epidemiology. 2012 Jan;23(1):1.

-This is a secondary analysis of existing data and is limited by the choices made in the initial study (which had the goal of studying fetal growth in healthy pregnancies). It was chosen for further analysis due to the availability of both environmental and stress markers, which is fairly unusual. Your concern for bias is true. We have added something more to that effect in the limitations: Dionisio KL, Baxter LK, Chang HH. An empirical assessment of exposure measurement error and effect attenuation in bipollutant epidemiologic models. Environ Health Perspect. 2014 Nov;122(11):1216-24. doi: 10.1289/ehp.1307772

-We have also highlighted that this study cannot be generalized.

The choice of statistical approaches needs to be better described and defended, and additional limitations need to be acknowledged. For example, while the authors acknowledge that there may be interactions and/or correlations between exposures, their first method, based on creating a cumulative risk score, does not allow for them to be explicitly modeled (as far as I can tell). While the demonstration of the WQS regression method is interesting and appropriate, more discussion of its limitations is warranted. For example, the WQS regression method underweights variables that have high pairwise correlation. In particular, the method can break down when there is high correlation between exposures that dominate the correlation with the outcome. This seems like it may be a possibility in this study, and the authors should consider how this may affect their findings. In addition, the method assumes a constant change in risk between the quantiles, and assumes the effects of all exposures operate in the same direction. The authors should improve their discussion of the limitations of the method, and acknowledge how the method’s limitations may affect their conclusions.

-All risk factors’ effects were in the same direction. Referring to Table 3, the exposures were not highly correlated.

Minor comments:

Introduction

The authors identify race as a risk factor for SGA; however, much evidence and theory points to racism, rather than race, as the mechanism that underlies documented associations between race and many adverse health outcomes. Historical policies and cultural forces that enforced economic exclusion and marginalization by race/ethnicity are also relevant when discussing the correlations between low income and educational attainment and race/ethnicity. These are important distinctions that the authors do acknowledge briefly but should mention each time they discuss race/ethnicity as an “exposure” or allude to the correlation of race/ethnicity with other risk factors. For example, the authors say in the introduction “Several possible risk factors for SGA have been identified, and include environmental toxicants, race, ethnicity, income, maternal educational attainment and maternal mental health…”. Instead they might consider “Several possible risk factors for SGA have been identified, and include environmental toxicants, structural and interpersonal racism, income, maternal educational attainment and maternal mental health…” See the following for more discussion on this topic: Krieger N. Does racism harm health? Did child abuse exist before 1962? On explicit questions, critical science, and current controversies: an ecosocial perspective. American journal of public health. 2003 Feb;93(2):194-9. Dominguez TP. Adverse birth outcomes in African American women: the social context of persistent reproductive disadvantage. Social work in public health. 2011 Jan 1;26(1):3-16.

-It is for this reason that we considered race as a separate variable, rather than incorporating it into a socioeconomic index as has been done in some analyses. A risk factor is not necessarily a cause, and part of the goal of this analysis is examining how disadvantage on multiple fronts may contribute to racial and other disparities. The conceptual model now added into the paper highlights this point. We have also made the correction to the text as you suggested.

Materials and methods

The authors claim that women were recruited if they did not consume alcohol and were nonsmokers. What is the relevant timeframe for these behaviors? Women who have never consumed alcohol and have never smoked are likely different from women who did not consume alcohol or smoke during pregnancy or in the year before birth. The authors should include a reference to the questionnaire that was used to evaluate these behaviors. They should also include a figure that illustrates how the final sample size was achieved. The authors cite a cohort profile that presumably goes into more detail about the study population and measures, but some additional information should be included in the manuscript. For example, are all the visits during pregnancy? The authors state there are 6 visits in the Study participants section, but then allude to 7 visits in the Variables section. Do the 5 study visits included in this study include the recruitment visit or not?

-The study included one recruitment visit and six follow-up visits. Due to lack of data in the sixth visit, we only used data from 5 follow-up visits.

When were maternal blood samples taken? A flow diagram of the study visits and timing of each measurement would be helpful.

-Added to section 2.3

What are the 12 locations where women were recruited? The effect of income on health may be very different depending on cost of living which varies widely by city and region in the US.

-Added to the manuscript

The authors claim they included values that were below the LOD in the analysis to prevent introducing bias. How was this done? This should be included in the methods.

-We used the values that were produced in the analysis

There are details relevant to the statistical analysis that are not clearly described enough. For example, the authors sum indicators of exposure to create a cumulative risk score, and then divide that sum into tertiles, with the highest tertile considered “exposed”. This would then assume each exposure has equal weight, which I am not sure makes sense.

-Cumulative risk scores are used in order to combine correlated exposures. Part of our motivation for conducting this study using 2 different methods was to observe whether the methods produced different results. We have included this goal in our objectives section.

The authors should discuss the implications of this and/or include it as a limitation. Was there any missing data and/or loss to follow-up in the study, and if so, how was that handled? A description of this should be included in the methods.

-A flow chart is now provided.

For the WQS regression, did the authors use p=0.10 as their significance threshold, or is that just an example?

-Yes, revised to reflect that we used p=0.10

If it was an example, what was the threshold they used? Did they test the sensitivity of their results to this threshold? What does it mean that the “weights are based on the quantiles of the scales”? What are the scales? Can the authors give more information as to how they “created weights adjusting for race/ethnicity and confounders”? Were these weights the weights used in the WQS regression?

-The weights were used in the WQS method. It is now more explicitly described (line).

Results

The paragraphs on either side of Table 4 on page 9 are repeated.

The results from the analysis using tertiles presented in Table 4 are not discussed in the text.

-Appears that there was a formatting error. It is now corrected.

The authors say that “the weight for education was 0.96 while it was only 0.04 for income.” In Table 5, however, the weights are 0.88 for education and 0.12 for income.

-It has been corrected

I am not sure where the numbers in the text come from. We caution the authors to rely heavily on statistical significance when interpreting their results. For example, in Tables 4 and 5 there are several point estimates of similar size, but only a few confidence intervals where the lower CI is below 1. Recent guidance from statisticians strongly proposes that these differences be discussed with more nuance rather than focusing solely on the results whose confidence intervals do not cross the null or where the p-values are less than 0.05. For discussion of this point of view, see the following commentary: https://www.nature.com/articles/d41586-019-00857-9?fbclid=IwAR1jzbGpWu9wsHIwBdOu3byOielCLEQxPZMvHJ-3X4GW2gvy4eD98a7a9EU

-This caution is noted. In the unadjusted domain, metals and psychosocial domains’ lower confidence interval was close to the null (0.97 and 0.96 respectively). However in adjusted models, the confidence interval tended to shift (as did demographics). This is noted in the results section now.

Discussion

It is not clear to me how this study “helps explain how lower levels of exposures jointly increase women’s risk for adverse birth outcomes”. It just seems like this index was dominated by the socioeconomic indicators, and the stress and chemical exposures had no relationship with SGA separately or jointly. Did the authors consider analyzing the stress and chemical exposures jointly, while adjusting for income and education? Or since income and education may influence stress and chemical exposures, did they examine whether they could observe those relationships in their study, beyond just looking at bivariate correlations?

-The study purpose was to assess multiple methods of dealing with this issue, incorporating multiple stressors on the chemical, social, and psychosocial level, theorizing that all domains might have effects. Models are presented that include the individual factors, mutually adjusted (reference table). This mutual adjustment had relatively little effect on the effect sizes. As indicated in the referenced paragraph as a whole, the paper presents multiple options for addressing this type of data. The cumulative index was associated with SGA, although further analysis indicated a strong effect of the socioeconomic factors.

Reviewer 3 Report

The authors studied two modeling techniques for measuring associations in a cumulative environmental risk framework. They explored how maternal chemical and nonchemical stressors may be associated with small for gestational age (SGA) among obese women enrolled in the NICHD Fetal Growth Studies. The authors grouped exposures into three areas or domains: metals (maternal lead, mercury, cadmium),  psychosocial stress (Cohen’s perceived stress, Edinburgh depression scores) and sociodemographic index (race/ethnicity, income, and education). The authors presented odds ratios using a cumulative index and Weighted Quantile Sum (WQS) regression.

My main comments are as follows:

1) This study is important because there is a major gap in the literature with few studies exploring cumulative risk frameworks examining both environmental and sociodemographic exposures and their associations with pregnancy outcomes. The study presents a high quality dataset in which to explore the hypotheses. The comparison of methods is informative, but the authors could do more to draw conclusions about the relative strengths/weaknesses of the two frameworks.

2) With the increasing prevalence of obesity and overweight, the study presents a novel population to study. However, the authors could strengthen the presentation by making the rationale for selecting obese women more evident and explaining this choice and better situating their work in the current literature considering susceptibility of this population to environmental risks (see examples in the air pollution context).

Woodruff TJ, Carlson A, Schwartz JM, Giudice LC. Proceedings of the summit on environmental challenges to reproductive health and fertility: executive summary. Fertil Steril. 2008;89:e1-e20.

Huang, H., Wang, A., Morello-Frosch, R., Lam, J., Sirota, M., Padula, A., & Woodruff, T. J. (2018, March 1). Cumulative Risk and Impact Modeling on Environmental Chemical and Social Stressors. Current Environmental Health Reports. NLM (Medline). https://doi.org/10.1007/s40572-018-0180-5

Koman PD, Mancuso P. Ozone exposure, cardiopulmonary health, and obesity: a substantive review. Chem Res Toxicol. June 2017:acs.chemrestox.7b00077. doi:10.1021/acs.chemrestox.7b00077

3) Regarding the methods and results, in general the methods are appropriate. The clarity of the methods used could be improved and the discussion of measurement error and confounding is not well developed and needs to be bolstered. In the Introduction, the authors present that educational attainment is a known independent factor associated with the outcome, and then without explanation or discussion of the possible limitations, the authors combine this factor into an index with another highly correlated factor. The statistical consequences of this technique in the modeling framework should be discussed. In addition, air pollutants are an unmeasured confounder that should be discussed (see point 2).

4) The authors present a tacit explanation that the source of the metals is industrial sites and other unnamed sources; however, the authors don’t test this hypothesis in their dataset in which the exposures to metals are relatively low. An alternative explanation is diet, commercial and consumer products, or occupational exposure that might not have a relationship to the other factors in the same way as settlement patterns/industrial zoning. The authors should more clearly state their assumptions, evidence, or claims about the sources of the environmental exposures, especially as it relates to the possible correlation among variables discussed and the implications for the two modeling frameworks.

5) Areas where a more nuanced presentation could strengthen the interpretation are listed below by line number.

Specific comments

Line 20 (abstract) – should mention that these are obese women and any other relevant population characteristics briefly.

Introduction: The authors should strengthen the presentation by making the rationale for selecting obese women more evident (see #2 above). Is there evidence for the risk factors among obese women and what is the direction of the association? What are the plausible biological mechanisms?

Methods: It would be helpful to have a little more information about WQS in the paper. I suggest adding in the equations for clarity (perhaps in a supplement if there isn't room in the manuscript).

Line 173: Specify the timeframe over which you hypothesize that these factors impact the outcome and the evidence that supports the timing of exposure being the same for all factors (or if that is an assumption, make that clearer).

Age is also associated with metals exposure and build-up in the body. How do these methods appropriately control for confounding by this factor?

Do these participants differ by access to pre-natal care? This should be described.

Discussion

Line 300: please specify the definition of vulnerability you are using. You may want to separately define vulnerability and susceptibility (and this might be more appropriate for the introduction).

Line 303- How did you test for this hypothesis in your dataset? (and for exposure to these specific metals at this type of environmental exposure level)?

Line 303 - Scientific "theory" is not the most precise term and it should be replaced.

Line 321 - add in "obese" for clarity.

322 - Please exlain why. Could this association be confounded by other unmeasured contaminants such as air pollution?

Line 330 - I was confused by this sentence - please re-write with attention to exposure v. risk.

Author Response

Dear Reviewer,

Thank you for your comments. We have carefully edited based on your suggestions. Please see below.

This study is important because there is a major gap in the literature with few studies exploring cumulative risk frameworks examining both environmental and sociodemographic exposures and their associations with pregnancy outcomes. The study presents a high quality dataset in which to explore the hypotheses. The comparison of methods is informative, but the authors could do more to draw conclusions about the relative strengths/weaknesses of the two frameworks.

This has been noted. We have added more to the conclusions to in order to better support our hypotheses. The conclusion now says the following:

This study demonstrates that risk factors may affect SGA cumulatively and introduces WQS as a possible tool to measure cumulative risk. WQS has been presented as a possible tool to measure risks in perinatal research, as well as presenting the possibility of using the method in studies involving chemical and nonchemical risk factors. The present analysis also supports previous research in the importance of socioeconomic factors on reproductive health outcomes. When researchers study cumulative risk, special attention should be placed on acknowledging the important role of sociodemographics on maternal health.

With the increasing prevalence of obesity and overweight, the study presents a novel population to study. However, the authors could strengthen the presentation by making the rationale for selecting obese women more evident and explaining this choice and better situating their work in the current literature considering susceptibility of this population to environmental risks (see examples in the air pollution context).

The study was a secondary analysis of a NICHD study. The study comprises all weight classifications. However, NICHD specifically oversampled for obese women in order to conduct analyses on fetal growth in women experiencing GDM. I have edited the explanation of this for more clarity in lines 131-135

Regarding the methods and results, in general the methods are appropriate. The clarity of the methods used could be improved and the discussion of measurement error and confounding is not well developed and needs to be bolstered.

The methods section has been edited for clarity. As the paper addresses multiple exposures simultaneously and did not find strong effects, confounding is unlikely to be a major problem. However, measurement error is probably a stronger issue, especially when the fact that multiple factors are being added together that include errors. We have added this to the discussion under limitations.

In the Introduction, the authors present that educational attainment is a known independent factor associated with the outcome, and then without explanation or discussion of the possible limitations, the authors combine this factor into an index with another highly correlated factor. The statistical consequences of this technique in the modeling framework should be discussed. In addition, air pollutants are an unmeasured confounder that should be discussed (see point 2).

Many of the exposure variables in environmental studies are highly correlated, as are social factors. The WQS method was designed to address this issue. We added another line that further illustrates this point (344-346). Creating an index, as we did in Method 1, is another method we used to account for the high correlation between variables. We have added air pollution as a potential confounder to the discussion (line 672).

The authors present a tacit explanation that the source of the metals is industrial sites and other unnamed sources; however, the authors don’t test this hypothesis in their dataset in which the exposures to metals are relatively low. An alternative explanation is diet, commercial and consumer products, or occupational exposure that might not have a relationship to the other factors in the same way as settlement patterns/industrial zoning. The authors should more clearly state their assumptions, evidence, or claims about the sources of the environmental exposures, especially as it relates to the possible correlation among variables discussed and the implications for the two modeling frameworks.

That is a very good point; we have added these issues to the background section (lines 163-165) as well as the discussion (633-639).

Specific comments

Line 20 (abstract) – should mention that these are obese women and any other relevant population characteristics briefly. Introduction: The authors should strengthen the presentation by making the rationale for selecting obese women more evident (see #2 above). Is there evidence for the risk factors among obese women and what is the direction of the association? What are the plausible biological mechanisms?

Please see above; the data comprises women of different weights, not just obese.

Methods: It would be helpful to have a little more information about WQS in the paper. I suggest adding in the equations for clarity (perhaps in a supplement if there isn't room in the manuscript).

Equations have been added to the manuscript

Line 173: Specify the timeframe over which you hypothesize that these factors impact the outcome and the evidence that supports the timing of exposure being the same for all factors (or if that is an assumption, make that clearer).

All exposures were measured during pregnancy; while in this case we were limited to data already collected, it seems reasonable to expect the strongest effects during this time frame.

Age is also associated with metals exposure and build-up in the body. How do these methods appropriately control for confounding by this factor?

All adjusted models include age.

Do these participants differ by access to pre-natal care? This should be described.

All women recruited were healthy and part of a cohort study that assessed exposures through prenatal care. Therefore, we were not able to assess this variable

Discussion

Line 300: please specify the definition of vulnerability you are using. You may want to separately define vulnerability and susceptibility (and this might be more appropriate for the introduction).

Included a clarifying sentence

Line 303- How did you test for this hypothesis in your dataset? (and for exposure to these specific metals at this type of environmental exposure level)? Scientific "theory" is not the most precise term and it should be replaced.

replaced by conclusion

Line 321 - add in "obese" for clarity.

Since not all are obese, we deleted “healthy”

322 - Please explain why. Could this association be confounded by other unmeasured contaminants such as air pollution?

Additional references have been added.

Line 330 - I was confused by this sentence - please re-write with attention to exposure v. risk.

Sentence has been reworded

Round 2

Reviewer 2 Report

Thank you for the improvements. However, there are several areas that need further explanation as outlined below.

Add racism as another box in the conceptual diagram, and have a link going into it from the race/ethnicity box. Then there should be links from racism to the social and physical environment box, the demographics box, and the psychosocial stressors box. The authors did not address whether the women never smoked and had never had alcohol, or whether they were current nonsmokers and nondrinkers at the time of recruitment. They also did not include a reference to the questionnaire. This should be addressed. The authors included a diagram of the sample size available for certain measures and the final data set size, but it’s not clear what variables were missing and how they arrived at the final data set size. They also did not address their rationale for doing a complete cases analysis versus multiple imputation or other missing data procedures. They should also add whether there was loss to follow-up, and if there was, why they chose not to implement inverse probability of censoring weighting or other approaches for addressing it.   The limitations of the statistical methods are still not described well enough. In response to the suggestion that WQS regression method can break down when there is high correlation between exposures that dominate the correlation with the outcome, the authors have stated that there is not strong correlation between the exposures. The issue, however, is if the correlation between the exposures is substantially stronger that the correlation with the outcome. The authors have not addressed this point. In addition, the method assumes a constant change in risk between the quantiles, which they have also not addressed. The also did not address the implications of the fact that they assume the cumulative risk score method weights each exposure equally. The authors need to improve their discussion of the limitations of both methods, and acknowledge how the method’s limitations may affect their conclusions.

Author Response

Dear Reviewer,

We have carefully considered your comments and have provided our response below. Please refer to the manuscript for our edits in response to your suggestions:

Add racism as another box in the conceptual diagram, and have a link going into it from the race/ethnicity box. Then there should be links from racism to the social and physical environment box, the demographics box, and the psychosocial stressors box.

We have now edited the first box in the diagram to say “factors related to race/ethnicity” and have included the phrase (e.g., racism) in order to more explicitly highlight this point.

The authors did not address whether the women never smoked and had never had alcohol, or whether they were current nonsmokers and nondrinkers at the time of recruitment. They also did not include a reference to the questionnaire. This should be addressed.

This is now addressed in lines 134-136. The study was limited to nonsmokers and nondrinkers. Women were retrospectively asked to report alcohol and smoking habits 3 months prior to pregnancy as well as current alcohol and smoking habits but the screening questionnaire was not provided with the data.

The authors included a diagram of the sample size available for certain measures and the final data set size, but it’s not clear what variables were missing and how they arrived at the final data set size. They also did not address their rationale for doing a complete cases analysis versus multiple imputation or other missing data procedures. They should also add whether there was loss to follow-up, and if there was, why they chose not to implement inverse probability of censoring weighting or other approaches for addressing it.  

One limitation of assessing cumulative risk is the that it will inherently cause a large amount of missing data when dealing with so many exposure variables (7 exposures in total were used). The diagram included demonstrates that n=524 women were missing information on at least one heavy metal. In addition, for psychosocial stress, at least n=285 women were missing at least one assessment of one psychosocial stress and n=336 women were missing at least one sociodemographic characteristic. For outcomes assessment, n=16 women were missing at least one data point either for gestational age or birthweight. Due to this amount of missing data, many of the methods to address missing data could not be employed. This is likely to create some bias in the results. We have added further discussion of this bias in lines 395-401.

The limitations of the statistical methods are still not described well enough. In response to the suggestion that WQS regression method can break down when there is high correlation between exposures that dominate the correlation with the outcome, the authors have stated that there is not strong correlation between the exposures. The issue, however, is if the correlation between the exposures is substantially stronger that the correlation with the outcome. The authors have not addressed this point.

As responded earlier, the reviewer’s point seems more of a theoretical concern, as the exposures in this study were not highly correlated. However, WQS was developed to address the problem of multiple correlated exposures (Carrico et al.) We have added a note that the performance of WQS is less good if exposure variables are strongly correlated with each other and weakly correlated with the outcome (Carrico).

In addition, the method assumes a constant change in risk between the quantiles, which they have also not addressed. The also did not address the implications of the fact that they assume the cumulative risk score method weights each exposure equally. The authors need to improve their discussion of the limitations of both methods, and acknowledge how the method’s limitations may affect their conclusions

The goal of the analysis was to compare a method that weights the exposure equally with a method that does not. This has been added to the introduction and the discussion, as well as possible non-linearity as a bias.

Reviewer 3 Report

The authors clarified many points and improved the presentation.

One change seemed to be detrimental.

I would strongly recommend that the authors remove Figure 1 as it does not fit with the correlation data presented by the authors in Table 3. I found the new figure was confusing and did not improve clarity - quite to the contrary.

One minor comment from the previous comment.

My previous comment: Age is also associated with metals exposure and build-up in the body. How do these methods appropriately control for confounding by this factor?

Author answer: All adjusted models include age.

My additional comment: This is an incomplete answer and misses my point. The equation does not distinguish the point I make. The authors might consider that age acts in two ways: 1) increases susceptibility and 2) increases exposure. How does controlling for one variable deal with these two factors?  

Author Response

Dear Reviewer,

We appreciate your comments and have provided our responses below. Please also refer to the manuscript for our edits in response to these suggestions

One change seemed to be detrimental. I would strongly recommend that the authors remove Figure 1 as it does not fit with the correlation data presented by the authors in Table 3. I found the new figure was confusing and did not improve clarity - quite to the contrary.

Reviewer one felt strongly that there was a need to explain how race and racism fit with the study exposures. We have left the figure in the manuscript as it helps to demonstrate the multicollinearity of exposures related to perinatal outcomes. We have included more explanation on why we included the diagram (lines 106-107).

My previous comment: Age is also associated with metals exposure and build-up in the body. How do these methods appropriately control for confounding by this factor? Author answer: All adjusted models include age. My additional comment: This is an incomplete answer and misses my point. The equation does not distinguish the point I make. The authors might consider that age acts in two ways: 1) increases susceptibility and 2) increases exposure. How does controlling for one variable deal with these two factors?

Accurately addressing this point would require detailed exposure measurements over time, to assess the age-exposure interaction and vulnerability. This was not possible with the type of data collected. We have added this as a limitation (lines 392-393) However, from Table 4, we can see that control for age only affected the association for sociodemographics. (Limited change in the odds ratios were seen in metals and psychosocial stress).